# Impact of Endothelial Progenitor Cells in the Vascularization of Osteogenic Scaffolds

**DOI:** 10.3390/cells11060926

**Published:** 2022-03-08

**Authors:** Dominik Steiner, Lea Reinhardt, Laura Fischer, Vanessa Popp, Carolin Körner, Carol I. Geppert, Tobias Bäuerle, Raymund E. Horch, Andreas Arkudas

**Affiliations:** 1Laboratory for Tissue Engineering and Regenerative Medicine, Department of Plastic and Hand Surgery, University Hospital of Erlangen, Friedrich-Alexander-Universität Erlangen-Nürnberg (FAU), 91054 Erlangen, Germany; reinhardt.lea@live.de (L.R.); laurafischer@berlin.de (L.F.); raymund.horch@uk-erlangen.de (R.E.H.); andreas.arkudas@uk-erlangen.de (A.A.); 2Preclinical Imaging Platform Erlangen (PIPE), Institute of Radiology, University Hospital of Erlangen, Friedrich-Alexander-Universität Erlangen-Nürnberg (FAU), 91054 Erlangen, Germany; vanessa.popp@uk-erlangen.de (V.P.); tobias.baeuerle@uk-erlangen.de (T.B.); 3Department of Materials Science and Engineering, Institute of Science and Technology of Metals, Friedrich-Alexander-Universität Erlangen-Nürnberg (FAU), 91058 Erlangen, Germany; carolin.koerner@fau.de; 4Institute of Pathology, University Hospital of Erlangen, Friedrich-Alexander-Universität Erlangen-Nürnberg (FAU), 91054 Erlangen, Germany; carol.geppert@uk-erlangen.de; 5Comprehensive Cancer Center Erlangen-EMN (CCC ER-EMN), 91054 Erlangen, Germany

**Keywords:** bone tissue engineering, endothelial progenitor cell, mesenchymal stem cell, vascularization, AV loop

## Abstract

The microvascular endothelial network plays an important role in osteogenesis, bone regeneration and bone tissue engineering. Endothelial progenitor cells (EPCs) display a high angiogenic and vasculogenic potential. The endothelialization of scaffolds with endothelial progenitor cells supports vascularization and tissue formation. In addition, EPCs enhance the osteogenic differentiation and bone formation of mesenchymal stem cells (MSCs). This study aimed to investigate the impact of EPCs on vascularization and bone formation of a hydroxyapatite (HA) and beta-tricalcium phosphate (ß-TCP)–fibrin scaffold. Three groups were designed: a scaffold-only group (A), a scaffold and EPC group (B), and a scaffold and EPC/MSC group (C). The HA/ß–TCP–fibrin scaffolds were placed in a porous titanium chamber permitting extrinsic vascularization from the surrounding tissue. Additionally, intrinsic vascularization was achieved by means of an arteriovenous loop (AV loop). After 12 weeks, the specimens were explanted and investigated by histology and CT. We were able to prove a strong scaffold vascularization in all groups. No differences regarding the vessel number and density were detected between the groups. Moreover, we were able to prove bone formation in the coimplantation group. Taken together, the AV loop is a powerful tool for vascularization which is independent from scaffold cellularization with endothelial progenitor cells’ prior implantation.

## 1. Introduction

Large-volume bone defects due to trauma or infection require reconstruction with autologous tissue. The transplantation of vascularized bone grafts such as the fibula or iliac crest is the current gold standard for the treatment of large bone defects [1,2]. However, the usage of such bone grafts can be associated with a considerable donor-side morbidity. Donor-side morbidity and/or comorbidities can limit the autologous tissue transfer [3]. The generation of tissue-engineered bone substitutes is a promising strategy to reduce the donor-side morbidity and to improve patient outcome. The engineering of bioartificial bone grafts requires (I) bone-forming cells, (II) osteogenic scaffolds and (III) sufficient vascularization. Mesenchymal stem cells (MSCs) represent an attractive cell source for tissue engineering applications. First described by Friedenstein et al. five decades ago, MSCs are multipotent progenitor cells with the ability to differentiate into mesenchymal tissues such as cartilage, fat, bone, muscle or tendon [4,5]. For bone tissue engineering applications, MSCs are widely used due to their easy isolation, high capacity for self-replication and osteogenic differentiation [6,7]. In the past, several biomaterials have been used as scaffold for bone tissue engineering applications. Hydrogels such as fibrin, alginate dialdehyde and gelatin (ADA-GEL) or recombinant spider silks display a good biocompatibility and biodegradation. Moreover, cells can be encapsulated into hydrogels and the high porosity supports nutrition and oxygen supply. In addition, the high water content and the ultrastructure enable an extracellular matrix-like milieu enabling structural support for cell transplantation [8,9,10,11,12,13,14]. Although different hard matrices for bone tissue engineering exist, we identified porous hydroxyapatite and beta-tricalcium phosphate granula (HA/ß–TCP) as a suitable osteogenic scaffold [15,16,17]. Sufficient vascularization is the fundamental limitation of tissue engineering applications. The integration of the tissue-engineered construct to the host vessel network ensures adequate oxygen and nutrition supply and thereby the survival of the transplanted cells. In the early postimplantation period, oxygen and nutrition supply is dependent on diffusion. Because diffusion is limited to 100–200 µm, different approaches have been performed to enhance scaffold vascularization. The so-called prefabrication is an encouraging methodology in order to vascularize tissue engineering constructs. Prefabrication can be achieved by two different microsurgical approaches: the arteriovenous bundle or the arteriovenous fistula (AV loop) [18,19]. The arteriovenous bundle is generated by the distal ligation of an artery with its venae comitantes. The arteriovenous fistula (AV loop) is formed by the anastomosis of a vein graft between an artery and vein. Tanaka et al. demonstrated that the AV loops display a higher potential for tissue generation and angiogenesis [20]. Another approach is the cellularization of the scaffold with endothelial progenitor cells prior to implantation. Endothelial progenitor cells display a high angiogenic and vasculogenic potential. In addition to that, EPCs can stimulate the osteogenic differentiation of MSCs in vitro and bone formation in vivo [21,22,23]. In our study we used the murine endothelial progenitor cell line T17b as described by Hatzopoulos et al. [24]. T17b EPCs are an exciting cell source for xenotransplantation studies since they do not express MHC I [25]. In several ischemia models T17b EPCs induced neovascularization and functional recovery of the ischemic tissues [26,27,28]. 

This study’s target was to investigate the influence of T17b EPCs on vascularization and bone formation of HA/ß–TCP–fibrin scaffolds. Moreover, this study analyzes the interplay between intrinsic vascularization originating from the AV loop and scaffold cellularization with T17b EPCs. 

## 2. Materials and Methods

### 2.1. MSC Isolation and Cultivation

The isolation and cultivation of mesenchymal stem cells (MSCs) from the bone marrow was performed according to an established protocol [10,29]. The Government of Mittelfranken and the Animal Care Committee of the University of Erlangen approved the procedure (55.2-2532.1-53/14). In brief, male Lewis rats were euthanized, the femur bones isolated and the bone marrow flushed with phosphate-buffered saline (PBS) and fetal calf serum (Biochrom, Berlin, Germany). After a centrifugation step, the cell pellet was reconstituted with DMEM (Gibco/Life Technologies, Carlsbad, CA, USA) containing 20% fetal bovine serum (Biochrom), 1% penicillin/streptomycin (Gibco Invitrogen) and 1% L-glutamine (Sigma-Aldrich, Schnelldorf, Germany). Then, the cells were filtered through 100 µm cell strainers (BD™, Becton Dickinson, Heidelberg, Germany) and density gradient centrifugation with Histopaque^®^ was performed (Sigma-Aldrich). Afterwards, the cells were cultured in cell culture flasks with a density of 2.0 × 10^6^/cm^2^ in a humidified atmosphere (37 °C; 5% CO_2_). The nonadherent cells were washed out after 48 h and the cell culture medium changed. MSCs were used until passage 5. 

### 2.2. T17b EPC Cultivation and Differentiation

The murine mesodermal endothelial progenitor cell line T17b was cultured and differentiated according to established protocols [24,30]. Briefly, T17b EPCs were seeded onto cell culture flasks coated with bovine skin gelatin type B (Sigma-Aldrich, Schnelldorf, Germany). For cell cultivation, high glucose DMEM GlutaMAX^®^ (Gibco/Life Technologies, Carlsbad, CA, USA) was used containing 20% fetal calf serum (Biochrom), 100 U/mL penicillin (Biochrom), 100 µg/mL streptomycin (Biochrom), 1 mM nonessential amino acids (Gibco), 2 mM HEPES buffer pH 7.5 (Gibco) and 0.1 mM 2-mercaptoethanol (Gibco). By supplementing 0.5 mM dibutyryl cyclic AMP and 1 µM all-trans retinoic acid (Sigma-Aldrich), endothelial differentiation was induced. 

### 2.3. AV Loop Operation 

The Animal Care Committee of the University of Erlangen and the Government of Mittelfranken approved the animal experiments (55.2-2532.1-53/14). Thirty male syngeneic Lewis rats (Charles River Laboratories, Sulzfeld, Germany) with a body weight between 240–440 g were used. Three experimental groups each with 10 animals were performed: matrix-only [group A], matrix and T17b EPCs [group B] or matrix and T17b EPCs as well as MSCs [group C]. As previously described, the arteriovenous loop operation was performed using an operating microscope (Carl Zeiss, Oberkochen, Germany) [8]. In brief, the saphenous vessels were dissected and a vein graft from the contralateral leg was anastomosed between the saphenous artery and vein, forming an arteriovenous loop (AV loop). Then, a porous titanium chamber was placed in the left groin and fixed onto the thigh musculature. Half of the matrix was filled into the chamber and the AV loop placed onto the matrix. The matrix consisted of porous hydroxyapatite (HA), beta-tricalcium phosphate (ß-TCP) granula (TricOs^®^, Baxter Healthcare, Vienna, Austria) and fibrin gel (TISSEEL Kit, Baxter Healthcare, Vienna, Austria). Prior to fibrin application, the thrombin and fibrinogen solutions were diluted with PBS in a 1:10 and 1:4 ratio. The diluted solutions were applied in a thrombin—fibrinogen ratio of 1:1 using the TISSEEL applicator. For cell implantation, T17b EPCs and MSCs were detached from cell culture dishes, centrifuged and the cell pellet reconstituted in fibrinogen. Group B contained 2 × 10^6^ T17b EPCs per construct. Group C contained 1 × 10^6^ MSCs and 1 x 10^6^ T17b EPCs per construct. After the AV loop was placed appropriately, the second half of the matrix was added and the titanium chamber closed with a lid (Figure 1A–D). Finally, the skin was closed.

### 2.4. Explantation Procedure

After 12 weeks, two rats per group were perfused with Microfil^®^ MV-122 solution (Flow Tech Inc., Carver, MA, USA) to perform CTs and the remaining animals received India ink solution. India ink solution consisted of 50% *v/v* India ink and 50% *v/v* Ringer solution containing 5% gelatine and 4% mannitol (Carl Roth, Karlruhe, Germany). As previously described, a median laparotomy was performed and the aorta was cannulated with a G21 cannula (Braun, Melsungen, Germany) [15,31]. After the caval vein was cut, the vascular system was flushed with 100 mL Ringer-Heparin solution (100 IU/mL) until clear fluid appeared. Then, 20 mL Microfil^®^ with 1 mL curing agent or India ink solution was applied, the caudal caval vein and aorta ligated and the specimen stored at 4 °C overnight. Thereafter, the constructs were explanted and fixed in Roti^®^-Histofix 4% (Carl Roth, Karlsruhe, Germany) for 24 h. For decalcification, the constructs were incubated in an ultrasonic bath (Bandelin sonocool Typ sc 255, Bandelin electronic GmbH und Co.KG, Berlin, Germany) with 20% EDTA solution (Sigma Aldrich, Steinheim, Germany) for 3 weeks. 

### 2.5. Computer Tomography

The microstructure of the newly formed vessel system was analyzed using computer tomography (CT) scans with an Inveon CT Scanner (Siemens Healthineers, Erlangen, Germany). The following scan parameters were applied: voltage of 80 kV, current of 500 µA, resolution of 24.49 µm per voxel and exposure time of 400 ms. Osirix Dicom Viewer (Aycan Osirix, New York, NY, USA) was used as imaging software.

### 2.6. Histological Staining

After decalcification, the constructs were embedded in paraffin and cut into 3 µm cross-sections, perpendicular to the longitudinal axis of the AV loop. Haematoxylin and eosin (H&E) as well as smooth muscle actin (α-SMA) staining were carried out according to standard protocols [32]. In order to prove potential immunogenic side effects of the osteogenic matrix, macrophages were visualized using CD68 staining. Briefly, the deparaffinized and rehydrated histological cross-sections were treated with a blocking solution (Zytomed Systems GmbH, Berlin, Germany). Thereafter, the anti-CD68 primary antibody (1:300 dilution, BIO-RAD, Hercules, CA, USA) was added and incubation occurred overnight. Finally, a second alkaline phosphatase-labeled anti-mouse antibody (AP-Polymer) and Fast Red TR/Naphthol AS (Sigma) substrate were added to induce the color reaction. Haemalaun was added for counterstaining.

To visualize matrix mineralization, alkaline phosphatase (ALP) staining was carried out. Briefly, the deparaffinized and rehydrated histological slices underwent a cooking step for antigen retrieval and a blocking solution (Zytomed Systems GmbH, Berlin, Germany) was added. Thereafter, the diluted (1:100) primary ALP antibody (GeneTex, Inc, Irvine, CA, USA) was added. After incubation at room temperature for 60 min, color reaction was induced with a second alkaline phosphatase-labeled anti-mouse antibody (AP-Polymer) and Fast Red TR/Naphthol AS (Sigma) substrate. Haemalaun was used for counterstaining.

Osteoclasts were detected using tartrate-resistant acid phosphatase staining (TRAP) according to an established protocol [33]. 

An Olympus IX81 microscope (Olympus, Hamburg, Germany), the Panoramic Flash scanner 250 and the software CaseViewer 2.4 (3DHISTECH, Budapest, Hungary) were used to take photographs of the histological cross-sections. Semiautomatic histological analysis was carried out according to established protocols [8,12].

### 2.7. Statistical Analysis

Statistical analysis was performed with GraphPad Prism 8.00 (GraphPad Software, San Diego, CA, USA). First, Shapiro–Wilk test was used for normal distribution. Then, statistically significant differences were calculated using an ordinary one-way ANOVA or Kruskal–Wallis test. *p*-values ≤ 0.05 were considered statistically significant. Results are shown as mean arbitrary units ± SD. 

## 3. Results

### 3.1. Surgical Outcome and Macroscopic Appearance

In group A, all animals survived the procedure. In group B and C, one animal died. Two animals (group A) displayed wound healing disorders and two constructs (group B and C) displayed signs of infection. The abovementioned four constructs have been excluded from further analysis. Additionally, one construct had to be excluded due to the extreme fragility of the construct. Six out of eight (75%; group A), four out of nine (44%, group B) and five out of nine (55%, group C) specimens displayed patent AV loops. After explantation, the constructs appeared dark-colored in the case of India ink perfusion or yellow-colored if Microfil^®^ was used for perfusion (Figure 2A,B). 

### 3.2. Biocompatibility and Degradation of the HA/ß–TCP–Fibrin Matrix 

CD68 staining was carried out to detect macrophages and multinuclear giant cells. In all three experimental groups, we were able to prove CD68-positive macrophages without multinuclear giant cells (Figure 3A,B). A quantitative analysis of CD68-positive cells revealed no statistically significant differences between the three experimental groups (Figure 3C). 

We used the construct weight and the histological cross-section area as surrogate parameters for biodegradation. After 12 weeks, the three experimental groups displayed no statistically significant differences considering the construct weight (0.52 ± 0.1 vs. 0.61 ± 0.01 vs. 0.62 ± 0.01 g; Figure 4A). In accordance with the construct weight, the histological cross-section area showed no statistically significant differences between the experimental groups (45.2 ± 9.9 vs. 38.2 ± 12.6 vs. 46.7 ± 8.4 mm^2^; Figure 4B). 

After 12 weeks, the fibrin gel was completely replaced by highly vascularized connective or bone tissue. The HA/ß–TCP granula displayed no complete degradation in all groups (Figure 5).

### 3.3. Vascularization and Bone Formation

As mentioned above, we were able to prove newly formed vessels originating from the AV loop in all constructs with patent AV loops. Interestingly, the transplantation of T17b EPCs and/or MSCs (group B and C) did not enhance the number of newly formed vessels compared to the matrix-only group (303 ± 196 vs. 422 ± 90 vs. 335 ± 71; Figure 6A). In addition to that, the vessel number per mm^2^ remained unaffected (7.9 ± 7.7 vs. 12 ± 4.3 vs. 7.2 ± 0.9; Figure 6B). Using α-SMA staining, we were able to prove that most of the newly formed vessels contained a media layer (Figure 6C,D).

Additionally, one specimen per group underwent computer tomography. The CT reconstructions revealed a dense vascular network originating from the AV loop (intrinsic vascularization) and from the periphery (extrinsic vascularization) (Figure 7A–C). 

We were able to prove incipient bone formation in the group containing T17b EPCs and MSCs (group C). Small parts of newly formed bone tissue were found close to the HA/ß–TCP granula and in the proximity of the vascularized construct parts (Figure 8A,B). Moreover, alkaline phosphatase (ALP) was found in the constructs containing T17b EPCs and MSCs (group C; Figure 8C). In addition to bone-forming cells, osteoclasts were also detected in group C using TRAP staining (Figure 8D). No bone formation and ALP activity as well as only a few osteoclasts were found in the groups without MSCs (groups A and B). 

## 4. Discussion

The integration of bioartificial tissues into the microvascular vessel network of the host organism is one of the major hurdles in the successful translation of tissue engineering applications into clinical practice. The implantation of endothelial progenitor cells into the scaffold is a promising strategy to promote vascularization. First described by Asahara et al. in 1997, endothelial progenitor cells display a high vasculogenic and angiogenic potential [34]. In previous studies, scaffolds containing EPCs displayed an increased vascularization compared to EPC-free ones [35,36,37]. However, when comparing the results of our study with other groups using cellularized scaffolds containing EPCs, it must be pointed out that these scaffolds were not additionally vascularized with AV loops to enhance vascularization and bone tissue formation. In our study, we were able to prove scaffold vascularization in all three groups after 12 weeks. In addition to intrinsic vascularization originating from the AV loop, we were able to prove extrinsic vascularization of the peripheral construct parts originating from the surrounding tissue. Although several studies demonstrated the high angiogenic potential of the T17b EPC cell line, we did not find a higher scaffold vascularization in the constructs containing T17b EPCs compared to the cell-free ones [26,38,39]. This result ties well with a previous AV loop study wherein the implantation of human umbilical vein endothelial cells (HUVECs) did not enhance scaffold vascularization [31]. Furthermore, the coimplantation of MSCs and T17b EPCs did not enhance scaffold vascularization, which is in contrast to the fact that MSCs can stimulate angiogenesis [9,36,40]. In a previous AV loop study, MSCs were encapsulated into alginate dialdehyde and gelatin (ADA-GEL) microcapsules and a Teflon chamber was used, allowing only intrinsic vascularization over a period of 4 weeks. This apparent contradiction might be explained by the long implantation period of 12 weeks and the usage of porous titanium chambers in the present study. It is alluring to speculate that MSCs and/or T17b EPCs might have positive effects on scaffold vascularization at an earlier time point, as the abovementioned previous AV loop study with a 4-week implantation period suggested [9]. Furthermore, it is possible that the combination of the extrinsic vascularization (by the usage of porous titanium chambers) and intrinsic vascularization (by means of AV loops) has a stronger effect on scaffold vascularization compared to the application of MSCs and/or T17b EPCs [17]. Nevertheless, one has to keep in mind that cell lines do not necessarily display the identical behavior in comparison to primary cells. Because of this potential limitation, future AV loop studies using primary EPCs are intended. Another approach to support vascularization and tissue formation independent from transplanted cells might be the use of exosomes derived from MSCs [41]. In this context, Hu et al. demonstrated that intramuscular injected exosomes promoted angiogenesis in a mouse ischemic hind limb model [42].

From the pertinent literature, it is well known that EPCs can stimulate the osteogenic differentiation of MSCs in vitro, as indicated by the increased expression of osteoblastic markers such as alkaline phosphatase (ALP) and matrix mineralization. Moreover, most studies recommend a cell ratio of 1:1 between MSCs and EPCs [23,43,44,45,46,47]. In our study, we also used a 1:1 cell ratio between T17b EPCs and MSC. We were able to show bone formation and alkaline phosphatase activity in the group containing MSCs and T17b EPCs, which is in line with previous in vivo studies from other groups [36,44,46]. Interestingly, no bone formation or alkaline phosphatase was detected in the T17b EPC-only group. In a previous AV loop study, we were able to prove bone formation in hydroxyapatite–fibrin scaffolds cellularized with HUVECs [31]. No bone formation or alkaline phosphatase were detected in the cell-free group, which is consistent with previous AV loop studies [31,48]. Furthermore, bone formation was lower compared to previous AV loop studies using MSCs or ADSCs [15,31]. In contrast to the abovementioned studies, the MSCs were not osteogenically differentiated prior to implantation, which might explain the lower bone formation. To circumvent this potential limitation, osteogenic-differentiated MSCs will be implanted in future bone tissue engineering studies using the AV loop. In addition to bone formation, osteoclasts were detected using TRAP staining in the proximity of the HA/ß–TCP granula as a proof of the remodeling processes of the hard matrix. Considering biodegradation, the HA/ß–TCP granula were not completely degraded after 12 weeks, in contrast to the fibrin gel. The latter one was completely replaced by highly vascularized tissue. The slow degradation of the HA/ß–TCP granula is a positive material characteristic because ideal scaffolds display an inverse correlation between biodegradation and (bone) tissue formation. Biocompatibility is an important characteristic of biomaterials. No signs of chronic inflammation have been verified in our study, demonstrating the good biocompatibility of fibrin and HA/ß–TCP. Although titanium nanoparticles released from medical implants are critically discussed as potential contributors to chronic inflammation, the animals tolerated the porous titanium chambers well [49]. On the other hand, innovative new biomaterials such as slow-degradable polycaprolactone coated with osteogenic molecules (e.g., ß-TCP) opens new perspectives for bone tissue engineering [50].

## 5. Conclusions

This AV loop study demonstrated the successful vascularization of an osteogenic HA/ß–TCP–fibrin scaffold. Bone formation was detected in the MSC/EPC coimplantation group, whereas no bone formation occurred in the T17b EPC-only and the matrix-only groups. In addition, the HA/ß–TCP–fibrin scaffold demonstrated a good biocompatibility and good structural support for bone formation in the coimplantation group.

## Figures and Tables

**Figure 1 cells-11-00926-f001:**
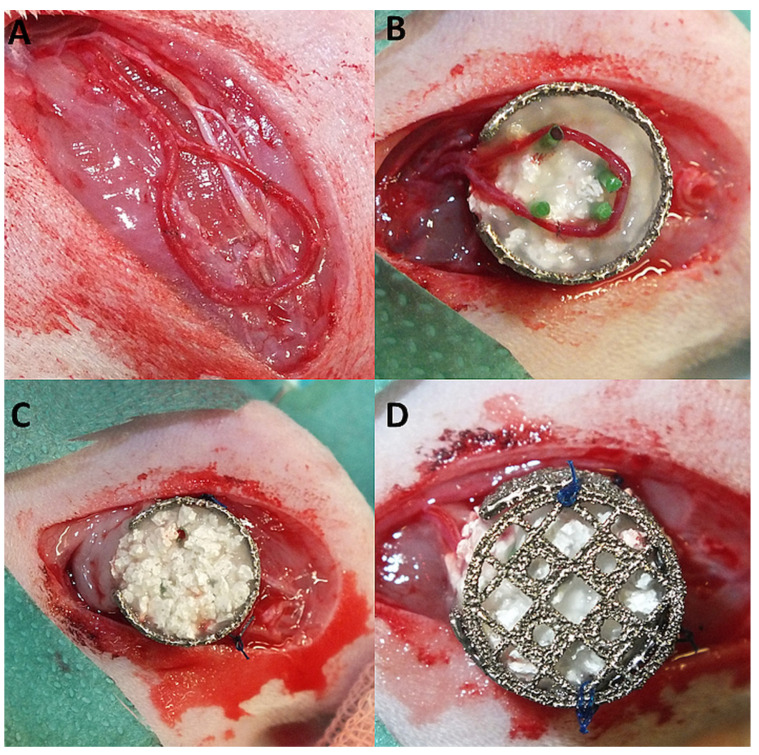
AV loop operation. First, the vein graft was microsurgically anastomosed between the saphenous vessels (**A**). Thereafter, the half-filled titanium porous chamber was sutured on the thigh musculature and the AV loop placed on top of the matrix (**B**). Finally, the second half of the matrix was added, the chamber closed with a lid and the skin closed (**C**,**D**).

**Figure 2 cells-11-00926-f002:**
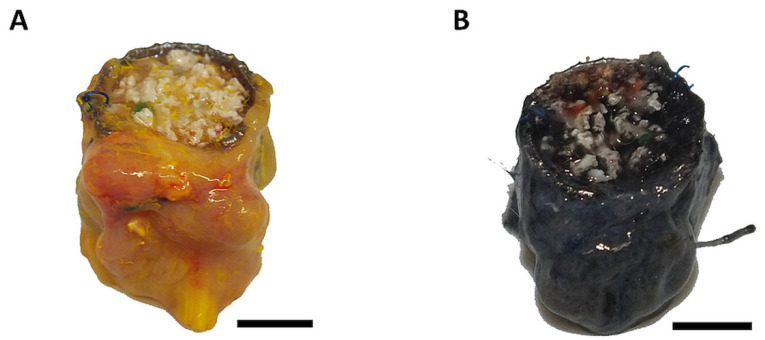
Macroscopic appearance. The explanted constructs appeared yellow-colored (**A**) if Microfil^®^ was used. In the case of India ink perfusion, the constructs appeared black (**B**).

**Figure 3 cells-11-00926-f003:**
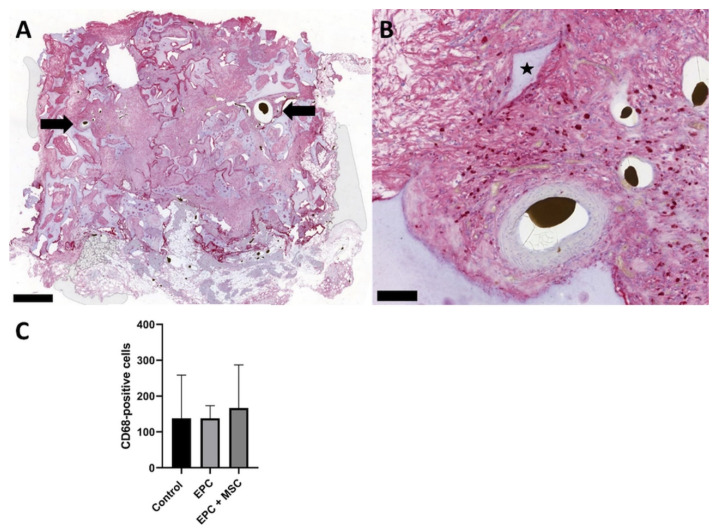
Macrophage staining. Using an anti-CD68 antibody, macrophages were visualized in the constructs. Macrophages are red-stained (**A**,**B**). Equal numbers of CD68-positive cells were found in all three groups (**C**). The cell-free group is indicated as “control”. Because no differences between the groups exist, exemplary histological slices from group B are demonstrated. Bold bars indicate the AV loop and the black star indicates the HA/ß–TCP granula. Scale bar = 1 mm (**A**) and 100 µm (**B**).

**Figure 4 cells-11-00926-f004:**
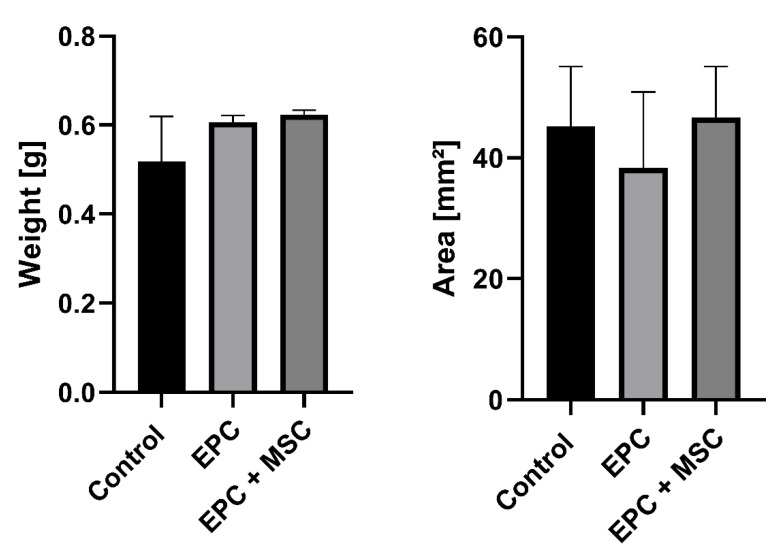
Construct weight and size. The experimental groups displayed no statistically significant differences considering the construct weight (**A**) and size (**B**). The cell-free group is indicated as “control”.

**Figure 5 cells-11-00926-f005:**
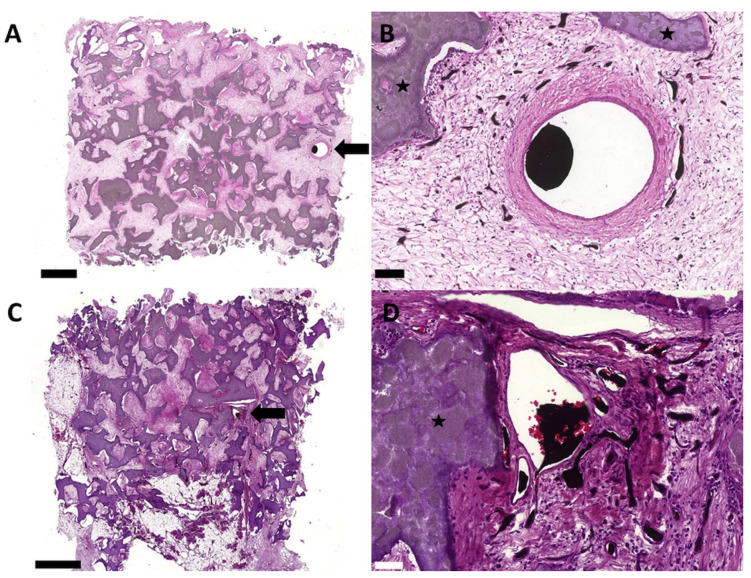
H&E staining with overview and detail images. The HA/ß–TCP granula were not completely degraded. The fibrin gel was replaced by highly vascularized tissue (**B**,**D**). No bone formation was observed in the cell-free (**A**,**B**) or T17b EPC group (**C**,**D**). Bold bars indicate the AV loop and the black stars indicate the HA/ß–TCP granula. Scale bar = 1 mm (**A**,**C**), 100 µm (**B**) and 50 µm (**D**).

**Figure 6 cells-11-00926-f006:**
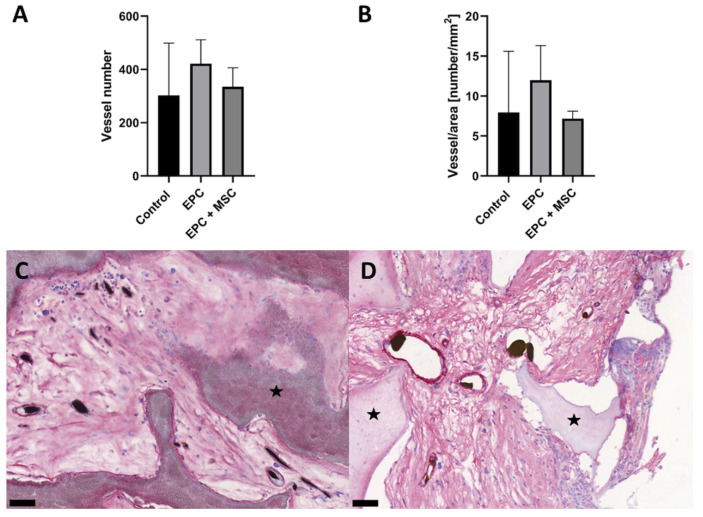
Quantification and detail view of vascularization. No statistically significant differences regarding the vessel number and vessel density were detected in the three groups (**A**,**B**). The newly formed vessels are filled with either Microfil^®^ or India ink solution and display a red-colored media layer in the α–SMA staining (**C**,**D**). Because no differences between the three groups exist, exemplary histological slices from group A and C are demonstrated (**C**,**D**). The black stars indicate the HA/ß–TCP granula. Scale bar = 50 µm.

**Figure 7 cells-11-00926-f007:**
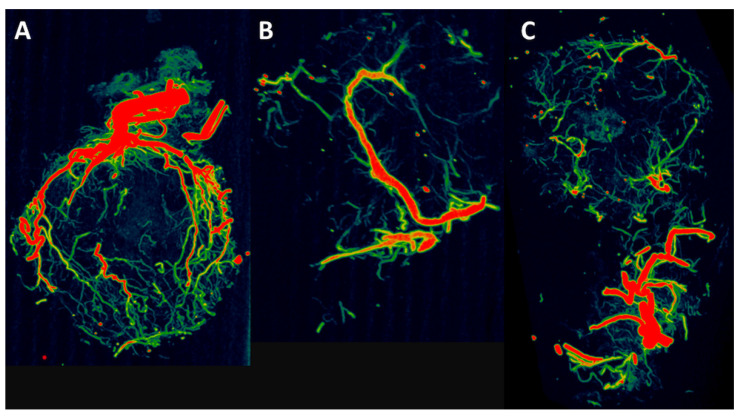
CT reconstructions. After 12 weeks one specimen per group was perfused with Microfil^®^ and underwent CT analysis. The CTs demonstrate the newly formed vascular network originating from the AV loop and from the periphery. (**A**) Cell-free group. (**B**) T17b EPC group. (**C**) T17b EPC and MSC coimplantation group.

**Figure 8 cells-11-00926-f008:**
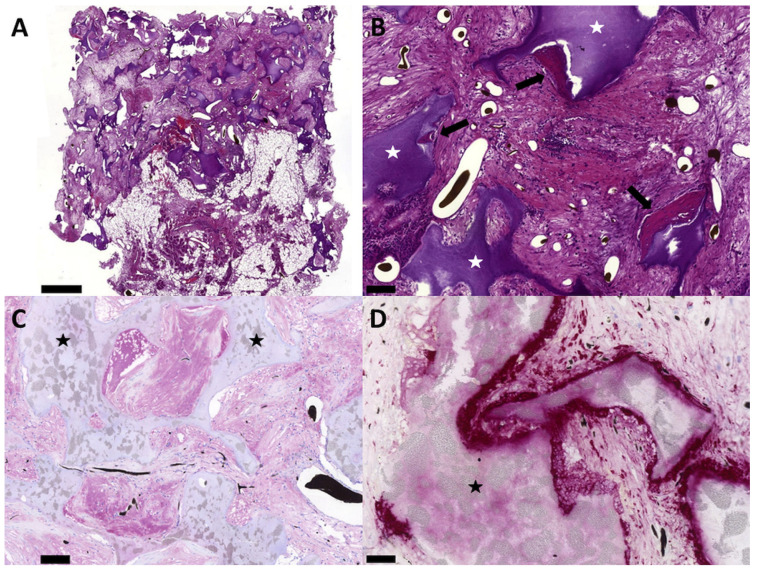
Bone formation, presence of alkaline phosphatase and osteoclasts. Only in the T17b EPC and MSC coimplantation group was bone formation detected (**A**,**B**). Bone formation occurred in the vascularized construct parts (**B**). Alkaline phosphatase was detected in the matrix in group C (purple-colored in **C**). Osteoclasts were found around the HA/ß–TCP granula in group C (red-colored in **D**). Bold bars indicate the newly formed bone tissue and the white/black stars indicate the HA/ß–TCP granula. Scale bar = 1 mm (**A**), 100 µm (**B**,**C**) and 50 µm (**D**).

## Data Availability

The data presented in this study are available on request from the corresponding author.

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
