# Peer review of "Impact of Endothelial Progenitor Cells in the Vascularization of Osteogenic Scaffolds"

_cells, 2022, doi:10.3390/cells11060926_

Round 1

Reviewer 1 Report

I congratulate the authors in this fine paper. Within tissue engineering, MSCs and EPCs are wiedly used, and showing the impact of their coimplantation in the clinically relevant A-V Loop model certainly mertis publication. There are some changes needed however, to make the paper better understandable. 

My biggest issue is the conclusion. In the abstact, the authors write: "...the AV loop in combination with the extrinsic vascularization mode is a powerful 33 tool for scaffold vascularization which is independent from additional endothelialization." I have trouble understanding the phrase extrinsic vascularisation. The A-V Loop chamber is closed off, meaning only the cells intrinsic to the construct can interact. So where is extrinsic vascularisation?This leads to the question of experiment design: why are the constructs closed off? Why did the authors not include a group without closing the construct off in a chamber? Not having a separation between the construct and the organism would be a more clinically relevant situation. This should be discussed.

Back to the conclusion: saying that vacularisation is independent of endothealization is an overstatement. The authors cocultivate with EPCs. We have no idea what these cells do in the construct. It might be something entirely different than epithealisation.

The cell line T17b is introduced (line 26) without explaining that this means.

The wording needs some improvement. E.g. line 43: "the latter one", or line 62 "auspicious methodology" (I don't believe that methodology can be auspicious, in the meaning of hopefull, or encouraging.

The experiment design and the results are well presented. After these corrections are done and the conclusions better explained, the paper can be published.

Author Response

Dear Bruce Gao,

Dear reviewer,

Thank you very much for the review of our original paper titled "Intrinsic vascularization of osteogenic scaffolds: impact of endothelial progenitor cells". We are very grateful for the helpful and detailed comments which significantly improved our manuscript. We have revised the manuscript as suggested and prepared it according to the guidelines for authors of the Journal “Cells”.

The enclosed document contains a revised version of the manuscript. Track change mode highlights the changes.

I would be delighted if you find the manuscript suitable in its current form for publication in “Cells”.

If you need any further information please do not hesitate to contact me.

Thank you very much for your efforts.

Yours faithfully

Dominik Steiner, M.D.

1. My biggest issue is the conclusion. In the abstact, the authors write: "...the AV loop in combination with the extrinsic vascularization mode is a powerful tool for scaffold vascularization which is independent from additional endothelialization." I have trouble understanding the phrase extrinsic vascularisation. The A-V Loop chamber is closed off, meaning only the cells intrinsic to the construct can interact. So where is extrinsic vascularisation?This leads to the question of experiment design: why are the constructs closed off? Why did the authors not include a group without closing the construct off in a chamber? Not having a separation between the construct and the organism would be a more clinically relevant situation. This should be discussed. Back to the conclusion: saying that vacularisation is independent of endothealization is an overstatement. The authors cocultivate with EPCs. We have no idea what these cells do in the construct. It might be something entirely different than epithealisation.

Answer: We thank the reviewer for this important comment. First of all we corrected the sentence in our abstract into: "Taken together, the AV loop is a powerful tool for vascularization which is independent from scaffold cellularization with endothelial progenitor cells prior implantation.

We have the impression that there might be a missunderstanding. We used a porous titanium chamber. This porous titanium chamber allowed also an extrinsic vascularization from the surrounding tissue.

2. The cell line T17b is introduced (line 26) without explaining that this means.

Answer: We've deleted "T17b" in the abstract and introduced the T17b EPC cell line in the introduction (line 80-83).

3. The wording needs some improvement. E.g. line 43: "the latter one", or line 62 "auspicious methodology" (I don't believe that methodology can be auspicious, in the meaning of hopefull, or encouraging.

Answer: We thank the reviewer for these helpful comments. We improved the wording.

Reviewer 2 Report

The authors have submitted “Intrinsic vascularization of osteogenic scaffolds: impact of endothelial progenitor cells”.

The authors need to perform the following changes:

  1. Title should be changed to “Impact of endothelial progenitor cells in the vascularization of osteogenic scaffolds”.
  2. Please explain if, and what kind of influence had the company producing “TricOs® –fibrin scaffold” on this research.
  3. Please AVOID commercial names (gel, ADA-GEL, as an example).
  4. The introduction is too short: you need to increase it.

This article is based on an interesting topic, well customized on the field of tissue engineering. The authors have well designed their starting hypothesis; however, I do suggest some improvements that may have impact also on the proper understanding of the main aspects of this paper.

Currently, a growing interest has been paid on extracellular environment and signalling, specifically, the research is interested on nanoparticles and exosomes: please discuss about it (See: Codispoti, B., Marrelli, M., Paduano, F., & Tatullo, M. (2018). NANOmetric BIO-Banked MSC-Derived Exosome (NANOBIOME) as a Novel Approach to Regenerative Medicine. Journal of clinical medicine, 7(10), 357.)

Authors have pushed their work on the role of angiogenesis and bone regeneration (see: Bressan, E., Ferroni, L., Gardin, C., Bellin, G., Sbricoli, L., Sivolella, S., Brunello, G., Schwartz-Arad, D., Mijiritsky, E., Penarrocha, M., Penarrocha, D., Taccioli, C., Tatullo, M., Piattelli, A., & Zavan, B. (2019). Metal Nanoparticles Released from Dental Implant Surfaces: Potential Contribution to Chronic Inflammation and Peri-Implant Bone Loss. Materials (Basel, Switzerland), 12(12), 2036.): in this landscape, a role could be played by inflammation and its triggers. Authors should briefly report how this could affect the overall impact of their research.

  • Discussion needs to be slightly increased.
  • Main limitations should be reported

Author Response

Dear Bruce Gao,

Dear reviewer,

Thank you very much for the review of our original paper titled " EIntrinsic vascularization of osteogenic scaffolds: impact of endothelial progenitor cells". We are very grateful for the helpful and detailed comments which significantly improved our manuscript. We have revised the manuscript as suggested and prepared it according to the guidelines for authors of the Journal “Cells”.

The enclosed document contains a revised version of the manuscript. Track change mode highlights the changes.

I would be delighted if you find the manuscript suitable in its current form for publication in “Cells”.

If you need any further information please do not hesitate to contact me.

Thank you very much for your efforts.

Yours faithfully

Dominik Steiner, M.D.

Title should be changed to “Impact of endothelial progenitor cells in the vascularization of osteogenic scaffolds”.

Answer: We thank the reviewer for this excellent suggestion. We've changed the title into "Impact of endothelial progenitor cells in the vascularization of osteogenic scaffolds"

2. Please explain if, and what kind of influence had the company producing “TricOs® –fibrin scaffold” on this research.

Answer: We're thankful for this important comment. The fibrin glue and HA/β-TCP matrix were provided by Baxter Healthcare but the company had no influence on our research. We've replaced TricOs into "HA/ß-TCP".

3. Please AVOID commercial names (gel, ADA-GEL, as an example).

Answer: We've deleted/replaced commercial names. Please be aware the "ADA-GEL" is the abbreviation of an hydrogel consisting of alginate dialdehyde and gelatin. We did not change "India ink" and "Microfil" in order to better separate the 2 different inks. 

4. The introduction is too short: you need to increase it.

Answer: Very good point. We provided more information concerning hydrogels (line 58-61) and the vascularization issue (line 67-68).

5.  Discussion needs to be slightly increased.

Answer: We increased the discussion section and added the paper "NANOmetric BIO-Banked MSC-Derived Exosome (NANOBIOME) as a Novel Approach to Regenerative Medicine" (307-310). Furthermore we added the second paper (line 334-341).

6. Main limitations should be reported.

Answer: We thank the reviewer for this important remark. From our point of the the cell line (line 304-307) and the non-steogenic differentiated MSCs (line 325-327) are the main limitations.

Round 2

Reviewer 2 Report

nothing to add